# Effect of Finishing Systems on Surface Roughness and Gloss of Full-Body Bulk-Fill Resin Composites

**DOI:** 10.3390/ma13245657

**Published:** 2020-12-11

**Authors:** Gaetano Paolone, Eugenio Moratti, Cecilia Goracci, Enrico Gherlone, Alessandro Vichi

**Affiliations:** 1Department of Dentistry, IRCCS San Raffaele Hospital and Dental School, Vita Salute University, 20132 Milan, Italy; euge_moratti92@hotmail.it (E.M.); gherlone.enrico@hsr.it (E.G.); 2Department of Medical Biotechnologies, University of Siena, 53100 Siena, Italy; cecilia.goracci@gmail.com; 3Dental Academy, University of Portsmouth, Portsmouth PO1 2QG, UK; alessandrovichi1@gmail.com

**Keywords:** bulk-fill, finishing, gloss, polishing, roughness

## Abstract

Background: In this study, we assess the effect produced on roughness and gloss of full-body bulk-fill materials by different finishing and polishing systems. Methods: Four full-body bulk-fill materials were tested: SonicFill2 (SF), Filtek Bulk Fill Posterior Restorative (FB), Tetric EvoCeram bulk-fill (EC), and Fill-Up! (FU). Sixty discs per material (2 mm in thickness and 7 mm in diameter) were obtained and randomly assigned (*n* = 15) to four finishing and polishing methods: Sof-Lex Spiral Wheels (SW), HiLusterPLUS (HL), Astropol (AP), and Opti1Step (OS). Surface roughness and gloss were then measured. Results: For roughness, material and surface treatment were significant factors (*p* < 0.001) with SF = FB = EC < FU and AP < SW < HL = OS. Material and surface treatment had a significant effect also on gloss (*p* < 0.001), with SF > FB = EC > FU and SW > AP > HL > OS. Conclusions: The tested combinations of bulk-fill and polishing systems provided clinically acceptable results with regard to roughness, while the outcome was poor for gloss. Multistep finishing/polishing systems were able to produce smoother surfaces on full-body bulk-fill materials compared to simplified ones.

## 1. Introduction

Resin composites (RCs) are widely used for anterior and posterior restorations, owing to their ability to mimic mechanical and optical properties of the natural tooth. As an improvement of early macro- and microfilled RCs, in the 1980s, “hybrid” RCs were developed. The initial limits in wearability and polishability were progressively overcome as a result of the increase in filler load and the reduction of filler size [1], leading to the development of microhybrid and then nanohybrid RCs. However, the increase in filler load also determined an increase in rigidity and curing stress, thus requesting the use of appropriate layering techniques. More recently, the advent of new low-stress monomers, highly reactive photoinitiators, and different types of nanosized fillers led to the introduction of new RC formulations, defined as “bulk-fill” materials, aimed to a single increment filling technique [2,3].

The first developed bulk-fill resin composites were meant as base materials to be capped with a conventional resin composite as the occlusal layer. This original category of bulk-fill composites has been classified as “base” [4] or “low viscosity” [5]. More recently, bulk-fill composites with a higher filler load have been introduced as one-step materials that do not require an occlusal capping. This newer category has been classified as “full-body” [4] or “high viscosity” [5]. This newer formulation has gained popularity among clinicians for its ease of use [3,4,5]. While several properties of full-body bulk-fill RC materials such as flexural strength, degree of conversion, elastic modulus, surface hardness, and monomer elution [6] have been investigated, scarce information is available regarding their finishing and polishing. Differences in filler type and size between full-body bulk-fill and traditional nanohybrid RCs, as well as among the various full-body bulk-fill RCs available, are expected to affect their polishability, also considering that no specific finishing and polishing procedure has been proposed for this class of materials and that finishing and polishing systems meant for conventional microhybrid RCs are generally used. Polishability is a pivotal property of RCs and surface characteristics such as roughness and gloss play an important role in determining clinical outcome of the restorations. Inadequately finished and polished surfaces are indeed more prone to wear and plaque accumulation, thus exposing the restored tooth to a higher risk of staining, secondary caries, and gingival irritation [7,8], possibly compromising clinical success [9,10,11]. Furthermore, it is well known that restorations with smooth surfaces are more comfortable, more aesthetically pleasing, and better accepted by patients [12,13].

The purpose of this study was then to evaluate surface roughness and gloss of full-body bulk-fill materials treated with different finishing and polishing systems that are not specifically developed for this category of materials. Two null hypotheses were formulated and subjected to test. The first null hypothesis was that neither the material type nor the performed finishing and polishing procedure had a significant influence on the surface roughness of currently available full-body bulk-fill RCs. The second null hypothesis was that surface gloss of these same RCs was not significantly affected either by the material type or by the followed finishing and polishing method.

## 2. Materials and Methods

Four full-body bulk-fill materials were selected for the study: SonicFill2 (SF) shade A2 (Kerr, Orange, CA, USA), Filtek Bulk Fill Posterior Restorative (FB) shade A2 (3M-ESPE, St. Paul, MN, USA), Ivoclar Tetric EvoCeram bulk-fill (EC) shade IVA (Ivoclar Vivadent, Schaan, Liechtenstein, and Fill-Up! (FU) shade Universal (Coltène/Whaledent, Altstätten, Switzerland. All these materials are adequate for placement in occlusal surfaces. The composition of the tested materials is reported in Table 1. For each material, 60 disc-shaped specimens of 7 mm in diameter and 2 mm in thickness were fabricated by placing the material into customized PTFE molds between two glass slides and polymerizing for 40 s with an LED curing light (Valo, Ultradent, South Jordan, UT, USA). The light-curing unit had an output of 1000 mW/cm^2^ and the light tip was placed at a standardized distance of 1 mm from the specimen surface. To allow for composite postcure, specimens were left in an incubator at 37 °C and 100% humidity for 24 h before use. A #1982M medium grit Sof-Lex disc (3M ESPE, St. Paul, MN, USA) was used for 20 s to create a standardized initial roughness on the specimen surface, simulating the one produced clinically by a red diamond bur. A new Sof-Lex disc was used for each specimen. Manual finishing and polishing procedures of all the specimens were then performed by one same operator (EM). The operator was calibrated using a precision scale before and during the procedure, considering that a 40 g force was considered light pressure. The operator calibration was repeated every 10 specimens [14]. Fifteen specimens of each RC were randomly assigned to one of the four finishing and polishing techniques as follows:

Group 1 (SW): Sof-Lex™ Spiral Wheels (3M ESPE, St. Paul, MN, USA), 20 s for each system step, using a low-speed handpiece in circular movements under continuous water cooling (50 mL/min);

Group 2 (HL): HiLusterPLUS Polishing System (Kerr, Orange, CA, USA) 20 s for each system step, using a low-speed handpiece in circular movements under continuous water cooling (50 mL/min);

Group 3 (AP): Astropol (Ivoclar Vivadent, Schaan, Lichtenstein) 15 s for each system step, using a low-speed handpiece in circular movements under continuous water cooling (50 mL/min);

Group 4 (OS): Opti1Step (Kerr, Orange, CA, USA) 20 s for each system, using a low-speed handpiece in circular movements under continuous water cooling (50 mL/min). Chemical composition and instructions for use of the finishing and polishing systems are reported in Table 2. Following finishing and polishing, all the specimens were rinsed thoroughly with distilled water and stored in the same medium for 24 h until roughness and gloss measurements were performed.

### 2.1. Surface Roughness

Before testing, specimens were ultrasonically cleaned in a 95% ethanol solution for 3 min. A profilometer (Mitutoyo SJ-201P, Mitutoyo, Kanagawa, Japan) set with a cutoff value of 0.8 mm, a stylus speed of 0.5 mm/s, and a tracking length of 5.0 mm was used [15] to assess surface roughness (Ra). The measurement setup was standardized by means of a custom mold for both the handpiece of the instrument and the specimen. Mean Ra (μm) was recorded.

### 2.2. Surface Gloss

The gloss measurements were performed using a small-area glossmeter (MG6- SA; KSJ, Quanzhou, China) with a square measurement area of 2 × 2 mm. Gloss assessment was performed at a 60° angle following the ISO 2813 specification for ceramic materials [16]. A black opaque plastic mold was placed over the specimen during measurements to avoid the influence of ambient light and maintain the exact position of the specimen relative to the glossmeter reading area.

### 2.3. Statistical Analysis

In order to test the first formulated null hypothesis, a two-way analysis of variance (ANOVA) was applied with roughness data as the dependent variable and material type and finishing/polishing system as factors, having preliminarily verified that roughness data were normally distributed (Shapiro–Wilk test) and that they had homogeneous group variances (Levene test). The Tukey test was applied for post hoc comparisons as needed.

The same statistical analysis (two-way ANOVA, Tukey test) was applied to gloss measurements to test the second formulated null hypothesis, once it had been checked that data distribution was normal (Shapiro–Wilk test) and that group variances were homogeneous (Levene test).

The statistical significance of the correlation between roughness and gloss measurements was assessed with the Pearson correlation test.

In all the statistical tests, the level of significance was set at *p* < 0.05. The SigmaPlot 11.0 software (Systat Software Inc., San Jose, CA, USA) was used for statistical calculations.

### 2.4. SEM Evaluation

Specimens’ preparation for SEM observations involved ultrasonically cleansing in a 95% alcohol solution for 3 min and air drying with an oil-free air spray. Specimens were then secured onto SEM (ZEISS EVO MA 10, ZEISS, Oberkochen, Germany) slabs with gold conducting tape and gold 80%/platinum 20% coated in a vacuum sputter coater (Quorum Q150R sputter coater, Quorum Technologies, Laughton, UK). The treated surfaces were then observed at 500× magnification (Figure 1).

## 3. Results

### 3.1. Surface Roughness

The descriptive statistics of surface roughness data are reported in Table 3. The two-way ANOVA showed that both material type (*p* < 0.001) and finishing/polishing system (*p* < 0.001) were significant factors for roughness. Particularly regarding material type, irrespective of the finishing/polishing procedure, the post hoc test highlighted that FU had a significantly higher roughness than the other materials (*p* < 0.05). Concerning the finishing/polishing system, it emerged from post hoc comparisons that, regardless of the material type, AP yielded the lowest roughness, and the difference was statistically significant (*p* < 0.05); SW produced a significantly lower roughness than HL and OS (*p* < 0.05).

### 3.2. Surface Gloss

The descriptive statistics of surface gloss data are reported in Table 4. According to the two-way ANOVA, both material type (*p* < 0.001) and finishing/polishing system (*p* < 0.001) were significant factors for gloss. When assessing the influence of material type per se, post hoc comparisons revealed that SF had the highest gloss, and the difference was statistically significant (*p* < 0.05). Furthermore, FB and EC exhibited a significantly higher gloss than FU. Concerning the finishing/polishing procedures per se, all the systems differed significantly according to the post hoc test (*p* < 0.05), with SW and OS yielding the highest and the lowest gloss, respectively.

The Pearson correlation test revealed that a moderate negative correlation existed between roughness and gloss measurements (Pearson correlation coefficient *r* = −0.534). The correlation was statistically significant (*p* < 0.001).

### 3.3. SEM Evaluation

Different surface topographies were observed for the different combinations of restorative materials and polishing systems. Filtek Bulk showed a quite smooth surface for all of the finishing systems tested, except for Opti1Step. Conversely, Fill-Up! showed a quite rough surface for all the finishing systems. More irregular results were observed for Evo Ceram and SonicFill 2 with some smooth combinations (SonicFill 2/Astropol, Evo Ceram/Sof-Lex Spirals) and other combinations with a higher degree of irregularities (Evo Ceram/Opti1Step, SonicFill 2/HiLuster). Most of the observed specimens showed homogeneous surfaces with few exceptions (Evo Ceram/HiLuster, SonicFill 2/Opti1Step).

## 4. Discussion

Numerous finishing and polishing systems for RC restorations are currently available and described in the literature [14,15,16,17,18,19]. While the effects of finishing and polishing on roughness and gloss of “conventional” hybrid and nanohybrid RCs have been largely investigated [9,10,11], little evidence has been collected thus far about the effects of finishing and polishing procedures on bulk-fill materials; recent studies have reported them to be 2–7 times rougher than nanofilled RCs [20,21,22,23]. In the present study, the effects of different finishing and polishing systems on roughness and gloss of different full-body bulk-fill materials were investigated. The collected data suggested rejection of the formulated null hypotheses that neither the material type nor the performed finishing and polishing procedure had a significant influence on surface roughness and gloss of full-body bulk-fill RCs.

It might be argued that a traditional hybrid resin composite should have been tested as the control group. This was indeed considered redundant, as the study was aimed at comparing different finishing/polishing systems on several available full-body bulk-fill materials and the comparative assessment of roughness and gloss was made with reference to threshold values of these properties that have been reported in the literature [24,25].

Roughness and gloss are clinically relevant characteristics of restorative materials. Roughness is related to irregularities and it is usually evaluated as roughness average (Ra), which is defined as the mean arithmetical value of all the absolute distances of the profile inside of the measuring length [26]. Gloss is an attribute of visual appearance that involves specular reflection from a surface; it is responsible for lustrous or mirror-like appearance [27,28] and it is measured in terms of gloss units (GU). Gloss is influenced by how light is reflected from the surface as well as by the refractive indices of resin matrix and filler [29]. Gloss was also found to be affected by filler size and filler–matrix homogeneity, with the following observation: the lower the filler–matrix homogeneity, the lower the light reflectivity [30]. When relating gloss to roughness in the present study, it emerged that an inverse linear relationship existed between the two properties. Such a finding is in line with the outcome of previous studies [22,31,32].

As the purpose of a finishing/polishing procedure is to provide enamel-like surfaces, ideally, the final composite roughness should be similar to enamel-to-enamel contact in occlusal areas (0.64 µm) [24]. Roughness data collected in the present study ranged from 0.11 µm to 0.69 µm. Therefore, all polishing systems provided clinically acceptable results. Nevertheless, statistically significant differences emerged among the materials. Particularly, FU had the highest Ra regardless of the finishing and polishing procedure. This finding could be related to the characteristics of the filler. FU indeed features the largest average filler size (2µm) among the tested materials. Moreover, Fill-Up! has a lower filler content (65 wt % or 49 vol %) compared to the other materials. However, as it is indicated for Class I and Class II restorations without the need of a covering, thus it can be considered a full-body, it was included in the study.

No statistically significant difference in roughness emerged among SF, FB, and EC. Such an observation is in line with the outcome of the study by O’Neill et al. [22], where no significant difference between SF and EC was recorded either in roughness or in gloss both before and after tooth-brushing simulation. Based on these findings and in accordance with Rigo et al. [23] and Attar et al. [17], the outcome in terms of surface roughness appears to be related to material composition.

The finishing/polishing procedures are also aimed at providing the restoration surface with an enamel-like gloss. Mormann et al. [33] reported 53 GU to be the reference value for the gloss of polished enamel, while Barucci-Pfister et al. [34] stated that the final gloss of a RC should be within the range of 40–53 GU. It should, however, be mentioned that no agreement has yet been reached in the literature on the geometry of viewing for gloss measurements, and the lack of uniformity in the experimental setup among different studies does not allow for a direct comparison of the published results. Some authors reported that a 20° angle enables a better differentiation than a 60° angle [35], while others reported the 45° angle as the best to detect between-material differences [34]. Cook and Thomas [25], using a 60° measurement angle, classified a finish below 60 GU as “poor”, a 60–70 GU finish as “acceptable”, and a finish above 80 GU as “excellent”. In the present study, the same 60° geometry, close to the surface observation angle of the surface by an average person, was selected, following the technical report of ISO 2813–2014 [16,36]. Therefore, according to the reported classification of Cook and Thomas, all of the polishing systems tested in the present study achieved a “poor” finish in terms of gloss.

The present study evaluated four finishing and polishing systems. Commonly utilized finishing and polishing systems involving the sequential use of a series of discs were not included, as they are not indicated for occlusal surfaces for which the bulk-fill materials of the present study are used. While bulk-fill materials enable a time-saving filling technique, there is still no consensus as to whether they also allow a simplified time-saving finishing/polishing procedure. In the present study, one-, two-, and three-step silicon points differed significantly from a two-step wheel system. The three-step system Astropol was found to provide the average smoothest surface compared to the other systems. The combination Sof-Lex™ Spiral Wheels+FB provided the smoothest surface. No differences were found between HiLuster and Opti1Step, being both significantly different from Astropol and Sof-Lex™ Spiral Wheels. Sof-Lex™ Spiral Wheels produced significantly higher gloss than the other investigated treatments. The combination Astropol+SF provided the highest gloss value among all the composites and polishing systems, whereas the combination 1Step+FU had the worst outcome for gloss. Sof-Lex™ Spiral Wheels provided consistent results on the different materials.

Irrespective of the number of steps, the chemical composition of the polishing points could account for the superiority of the three-step systems, considering that Astropol is the only one containing silicon carbide while the other two-step systems all have aluminum oxide.

As a limitation of the present study and as possible matter for future investigations, it should be mentioned that only one initial finishing system was used to create a standard surface for the present study. Nevertheless, several other finishing systems, such as diamond burs and carbide burs, are available, and their use can affect the initial roughness [37]. Stability of the initially achieved results over time (e.g., after function and/or brushing) could also be the object of a further research study, as scarce information is available on this issue.

## 5. Conclusions

Within the limitations of this in vitro study, the following conclusions can be drawn:

The combination of tested bulk-fill and polishing systems provided clinically acceptable results in terms of roughness, while poor results were achieved in terms of gloss.

The multistep finishing/polishing systems tested in the present study showed a higher polishing ability on full-body bulk-fill resin composite than the single step systems.

The study suggested the need for developing a polishing system dedicated to full-body bulk-fill resin composite.

## Figures and Tables

**Figure 1 materials-13-05657-f001:**
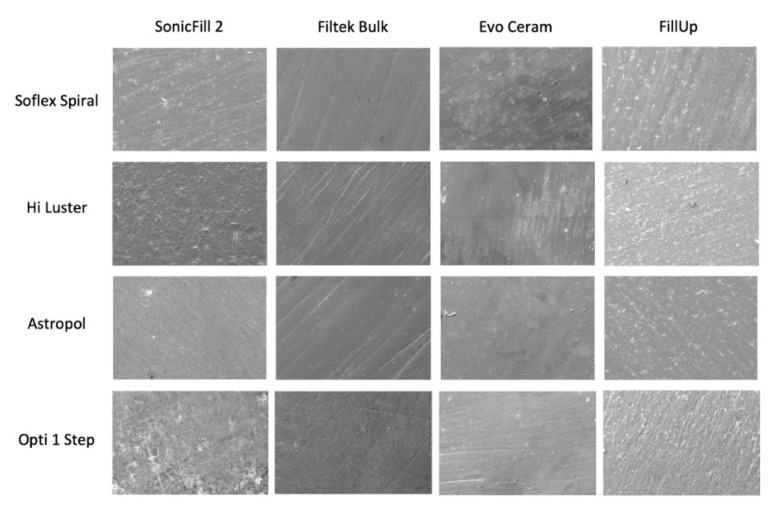
Scanning electron micrograph at 500× of the combinations of full-body bulk-fill and finishing systems tested.

**Table 1 materials-13-05657-t001:** Chemical composition of the tested bulk-fill resin composite systems. n.a.—not available.

Materials	Type	Composition	Filler Amount, wt %, vol %	Manufacturer
SonicFill2 (SF)	high viscosity light-cured bulk-fill material	Dimethacrylates, BisGMA, Bis EMA, Silica, Barium glass, YbF3, mixed oxides. Particle size 4μ.	81.35–n.a.	Kerr Corporation, Orange, CA, USA
Filtek Bulk Fill Posterior Restorative (FB)	high viscosity light-cured bulk-fill material	Matrix: AUDMA, UDMA, 12-dodecane-DMA (DDDMA); Fillers: nonagglomerated silica filler (20 nm), nonagglomerated zirconia filler (4–11 nm), aggregated zirconia/silica cluster filler (20 nm silica/4–11 nm zirconia), and an agglomerate ytterbium trifluoride filler (100 nm)	76.5–58.4	3M/ESPE, St. Paul, MN, USA
Tetric EvoCeram bulk-fill (EC)	high viscosity light-cured bulk-fill material	Dimethacrylates: Bis-GMA, Bis-EMA, UDMA, Barium aluminum silicate glass filler of a mean particle size of 0.4–0.7 μm; ytterbium fluoride and mixed oxides of a mean particle size of 160–200 nm; Ivocerin, a germanium-based initiator and a special shrinkage stress reliever	81–61	Ivoclar Vivadent, Schaan, Liechtenstein
Fill-Up! (FU)	medium viscosity dual-cured bulk-fill material	TMPTMA, UDMA, bis-GMA, TEGDMA, dibenzoyl peroxide; benzoyl peroxide, dental glass, Zinc oxide coated. Filler size 2μ.	65–49	Coltène/Whaledent, Altstätten, Switzerland

**Table 2 materials-13-05657-t002:** Chemical composition and instructions for use of the tested finishing/polishing systems. n.a.—not available.

Material	Type	Composition	Application	
Sof-Lex™ Spiral Wheels (SW)	2 steps	spiral finishing and polishing wheels—thermoplastic elastomer impregnated with aluminum oxide particles (pink and white)	20 s each, using a low-speed handpiece in circular movements and with continuous water cooling (50 mL/min)	3M/ESPE, St. Paul, MN, USA
HiLusterPLUS Polishing System (HL)	2 steps	Points with integrated aluminum oxide particles (1st step) and diamond particles integrated (2nd step)	20 s each, using a low-speed handpiece in circular movements and with continuous water cooling (50 mL/min)	Kerr Corporation, Orange, CA, USA
Astropol (AP)	3 steps	Silicon carbide–coated polishing points (coarse grey [F] 45 μm; fine green [P] 1 μm; and extrafine pink [HP] 0.3 μm)	15 s each, using a low-speed handpiece in circular movements and with continuous water cooling (50 mL/min)	Ivoclar Vivadent, Schaan, Liechtenstein
Opti1Step (OS)	1 step	n.a.	20 s each, using a low-speed handpiece in circular movements and with continuous water cooling (50 mL/min)	Kerr Corporation, Orange, CA, USA

**Table 3 materials-13-05657-t003:** Descriptive statistics of surface roughness (Ra, μm). Lowercase letters label statistically significant differences among finishing systems within each restorative material. Upper case letters label the statistically significant differences among the four materials within each treatment (*p* < 0.05).

Treatment	Roughness Ra (µm)											
	SonicFill 2			Filtek Bulk			Evo Ceram			Fill-Up!		
	Mean	SD	Sig.	Mean	SD	Sig.	Mean	SD	Sig.	Mean	SD	Sig.
Sof-Lex Spiral	0.14 ^A^	0.06	a	0.11 ^A^	0.04	a	0.18 ^A^	0.06	a	0.50 ^B^	0.13	b
HiLuster	0.24 ^AB^	0.07	b	0.20 ^A^	0.06	b	0.32 ^B^	0.07	b	0.69 ^C^	0.15	c
Astropol	0.13 ^A^	0.04	a	0.19 ^AB^	0.06	ab	0.15 ^A^	0.06	a	0.24 ^A^	0.08	a
Opti1Step	0.31 ^A^	0.08	b	0.30 ^A^	0.16	c	0.29 ^A^	0.08	b	0.50 ^B^	0.13	b

**Table 4 materials-13-05657-t004:** Descriptive statistics of gloss (GU). Uppercase letters label statistically significant differences among finishing systems per se. Lowercase letters label statistically significant differences among finishing systems within each restorative material. Uppercase letters label the statistically significant differences among the four materials within each treatment (*p* < 0.05).

Treatment	Gloss (GU)											
	SonicFill 2			Filtek Bulk			Evo Ceram			Fill-Up!		
	Mean	SD	Sig.	Mean	SD	Sig.	Mean	SD	Sig.	Mean	SD	Sig.
Sof-Lex Spiral	54.9 ^A^	15.4	a	45.0 ^BC^	11.3	a	38.4 ^C^	11.6	a	52.6 ^AΒ^	9.0	a
HiLuster	43.3 ^A^	12.0	b	42.1 ^A^	14.6	a	45.5 ^A^	9.8	a	3.1 ^Β^	1.12	c
Astropol	55.4 ^A^	13.7	a	44.1 ^B^	7.7	a	37.9 ^Β^	7.8	a	21.1 ^C^	12.0	b
Opti1Step	30.1 ^B^	5.5	c	28.9 ^B^	5.2	b	36.6 ^A^	6.2	a	6.6 ^C^	1.2	c

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
