# Peer review of "Effect of Finishing Systems on Surface Roughness and Gloss of Full-Body Bulk-Fill Resin Composites"

_materials, 2020, doi:10.3390/ma13245657_

Round 1
Reviewer 1 Report
I think that the article is well organized. However, line 41 "its ease of use...3.3.5" should be checked whether it is right form of citation.
Reviewer 2 Report
The authors reported surface roughness and gloss measurements of the combinations of four composites and four finishing/polishing systems. The whole design is just oversimplified and it is far from a systematic study. The four composites chosen in this study, although they are all bulk-filled composites, their compositions are very different, especially for the filler part in terms of loading and size, type of filler. All these parameters will affect the surface roughness and gloss values. The chosen of these four composites seems arbitrary. Overall, this study provides little novel scientific insights in dental materials, neither did it provide a useful reference for clinical practice.
(1) Each finishing/polishing system has its manufacture suggested composite types. For example, Astropol (P) is suggested for micro-filled composites while Astropol (HP) is suggested for hybrid composites. Therefore Astropol may be the best choice for FillUp!.
(2) For optimal roughness and gloss results, besides the choice of finishing/polishing, other parameters like application time and applied pressure are very important. This study fixed the pressure at 40g force, which is relatively small and may not be the optimal pressure for every composite-finishing/polishing combination. In order to provide more useful directions for dentists, the authors should all include the effect of pressure and application time into this study. By the way, can the authors explain why the application time for Astropol is 15 s, while for others are all 20 s?
(3) Instead of curing all the composites under same condition (1000mW/cm2 Valo LED light for 40 sec), composites should be cured by following their individual procedures required by their manufactures. For example, Fill-Up! is featured as fast-cure and only need 5 sec light irradiations under 1600mW/cm2.
(4) The arrangement of SEM images in Figure 1 should be in line with those in Table 3 and Table 4.
(5) Fill-Up! is not really a high viscosity RC. Its filler content (65wt% or 49vl%) is significantly lower than the other three composites in this study. Moreover, unlike the other three composites, Fill-Up! contains micro-size fillers. Therefore it is not reasonable to compare it with others.
(6) The authors did not mention the shades of the composites used in this study.
(7) Roughness and SEM samples were prepared by sonication in alcohol for 3min. Sonication in alcohol can lead to leaching of unreacted monomers and thus significantly affect the surface morphology, and affect the measured roughness value.
(8) Data quality of SEM images is low. It might not be easy for the readers to acquire useful information from these SEM images.
(9) The conclusion drawn by the authors are not solid. For example, one of the major claims made in this paper is “multi-step finishing/polishing systems have "better" polishing ability on high viscosity bulk”. This conclusion is misleading since the authors only studied one single-step finishing/polishing system.
Reviewer 3 Report
Dear Authors
The general meaning of the work is clear to me, but I am not sure if this work brings enough scientific input to be a publication in a good journal. From my point of view, it is rather a technical report devoted to examining the quality of polishing of various materials by various means.
I have a few reservations and substantive questions:
1. Why SEM photos have such low contrast? You can hardly see anything on them. I only needed 2 minutes to increase the contrast and make the micrographs easier to read. I add the effect as an attachment. I suggest improving the quality of Fig. 1
2. The profilometer used has several different measuring detectors in the manufacturer's catalog. Some of them are designed for materials more delicate than metals (e.g. polymers). What probe was used?
3. Why the roughness determination was not used with more scientific instruments that analyze the surface roughness, and not just along the lines? I am thinking of laser scanning digital microscope or AFM. There are also profilometers focused on scientific measurements, and not workshop ones, eg Dektak family.

Round 2
Reviewer 2 Report
The authors made some improvements in the revised manuscript. They provided SEM images with higher resolution and added some comments. I appreciate their efforts. However, their overall experimental design is still flawed and lacks novelty. The results and conclusions contribute little to peer researchers or dental practitioners. Fill-Up! is not really a high viscosity composite so it should not be included in this study. Additional detailed experiments may be necessary to make this study more complete and to add scientific significance required for acceptance. Suggested additional experiments may include but not limited to:
(1) As authors insisted in the reply that "Curing the composites under the same conditions is a standard procedure", they should at least provide the degree of vinyl conversion values of each sample after light-curing.
(2) Change of other properties such as color and surface hardness.
